# Peer reviews of peer reviews: A randomized controlled trial and other experiments

Alexander Goldberg [1]☉, Ivan Stelmakh[2]☉, Kyunghyun Cho[3], Alice Oh[4], Alekh Agarwal[5], Danielle Belgrave[6], Nihar B. Shah[1]

**1** School of Computer Science, Carnegie Mellon University, Pittsburgh, Pennsylvania, United States of America, **2** New Economic School, Moscow, Russia, **3** Center for Data Science, New York University, New York, New York, United States of America, **4** School of Computing, KAIST, Daejeon, Republic of Korea, **5** Google Research, United States of America, **6** GSK, London, United Kingdom

☉ These authors contributed equally to this work.

**Data availability statement:** We cannot share the raw data behind our results as we collected human subject data by asking participants in a peer review conference to evaluate the quality of (actual) paper reviews at the conference. This

## Abstract

Is it possible to reliably evaluate the quality of peer reviews? We study this question driven by two primary motivations – incentivizing high-quality reviewing using assessed quality of reviews and measuring changes to review quality in experiments. We conduct a large scale study at the NeurIPS 2022 conference, a top-tier conference in machine learning, in which we invited (meta)-reviewers and authors to voluntarily evaluate reviews given to submitted papers. First, we conduct a randomized controlled trial to examine bias due to the length of reviews. We generate elongated versions of reviews by adding substantial amounts of non-informative content. Participants in the control group evaluate the original reviews, whereas participants in the experimental group evaluate the artificially lengthened versions. We find that lengthened reviews are scored (statistically significantly) higher quality than the original reviews. Additionally, in analysis of observational data we find that authors are positively biased towards reviews recommending acceptance of their own papers, even after controlling for confounders of review length, quality, and different numbers of papers per author. We also measure disagreement rates between multiple evaluations of the same review of 28%–32%, which is comparable to that of paper reviewers at NeurIPS. Further, we assess the amount of miscalibration of evaluators of reviews using a linear model of quality scores and find that it is similar to estimates of miscalibration of paper reviewers at NeurIPS. Finally, we estimate the amount of variability in subjective opinions around how to map individual criteria to overall scores of review quality and find that it is roughly the same as that in the review of papers. Our results suggest that the various problems that exist in reviews of papers – inconsistency, bias towards irrelevant factors, miscalibration, subjectivity – also arise in reviewing of reviews.

data is potentially sensitive as participants in our study could have concerns around paper reviewers whom they deemed low quality learning the identity of their evaluator. In order to ensure privacy of participants, in our consent forms, we guaranteed participants in the study that "In the future, once we have removed all identifiable information from your data, we may use the data for our future research studies. The data will not be distributed, and only aggregate findings may be published." Therefore, we have provided partially aggregated data that allows researchers to further analyze each of our main results. The data is available in a public repository at OSF accessible here: https://osf.io/m5dch/ (https://doi.org/10.17605/OSF.IO/M5DCH).

**Funding:** The author(s) received no specific funding for this work.

**Competing interests:** The authors have declared that no competing interests exist.

## 1 Introduction

Scientific peer review is a ubiquitous process used across many fields to evaluate research quality. While the peer review of papers is widespread, it is plagued with well-documented problems like bias, subjectivity, fraud, miscalibration, and low effort, among others (see [1] for a survey). Some of these problems may be mitigated via design of better incentives for high-quality reviewing, or via evidence-based policy design evaluated through controlled experiments. Both of these approaches depend on reliable evaluations of the quality of reviews. Therefore, in this work we study the research question: can different parties involved in the peer-review process (meta-reviewers, reviewers, authors) reliably evaluate the quality of reviews? We are driven by the two primary motivations:

1. *Designing incentive mechanisms for high-quality reviewing.* A number of past works propose incentive mechanisms for the peer-review process to motivate better reviewing [2–6]. For example, reviewers may earn credit towards future peer review of their own work when they complete high-quality peer reviews of other's work. Already, at a number of journals and conferences, reviewers can be recognized for excellence in reviewing (e.g., NeurIPS "Top Reviewers") where this recognition is generally given out on the basis of evaluations of review quality completed by editors or meta-reviewers. The European Science Foundation reports that many grant organizations evaluate quality of reviews and a substantial fraction of these organizations store the evaluations linked to the reviewers' identities in their databases [7]. These mechanisms generally require reliable evaluation of review quality in order for incentives to be fair and useful. For example, [2] and [3] assume that authors will accurately provide a report of true review quality that can be used to incentivize effort on the part of reviewers.

2. *Experiments measuring efficacy of interventions in the peer-review process.* In numerous studies examining scientific peer review, the efficacy of changes to the peer-review process is assessed based on evaluating the quality of reviews under certain policy interventions (e.g., [8–18]). These studies treat evaluations of review quality by fellow reviewers or editors/meta-reviewers as "gold standard" to measure the efficacy of the policies under consideration. In our research, we delve into the validity of using these scores by examining their reliability as true indicators of review quality.

Motivated by the need for evaluations of review quality, we conducted a quantitative study into the reliability of evaluating review quality at the Neural Information Processing Systems (NeurIPS) 2022 conference, a top-tier conference in the field of machine learning. In computer science, unlike many other research fields, conferences typically review full papers, are frequently a terminal publication venue and are ranked higher than journals. We recruited participants who served in different roles in the conference — paper authors, paper reviewers, and meta-reviewers who handle many papers at the conference. We then asked these participants to evaluate the quality of paper reviews and analyzed the reliability of their scores in several ways. Additionally, we conducted a randomized control trial to examine potential bias in scores of perceived quality towards longer reviews.

Using the data collected we assess the reliability of evaluating review quality along five dimensions: (i) uselessly elongated review bias, (ii) author-outcome bias, (iii) inter-evaluator agreement, (iv) miscalibration, and (v) subjectivity. Overall, our findings suggest that the evaluation of paper reviews faces many of the same issues as the reviewing of paper quality,

like inconsistency, miscalibration, subjectivity, and biases with respect to irrelevant information. Therefore, care must be taken in relying on evaluation scores to either incentivize quality peer review or to experimentally measure changes in the quality of review due to these observed effects in evaluating review quality.

## 1.1 Related work

We discuss previous works that have conducted surveys of either authors, reviewers, or journal editors in order to study perceptions of review quality.

At the computer vision conference CVPR 2012, a study [19] asked paper authors to evaluate reviewer quality. They found that length had a weak positive correlation with author's ratings of "helpfulness." However, importantly, it is not possible to distinguish how much of the correlation was due to longer reviews having truly higher quality content versus longer reviews being spuriously perceived as higher quality. Our work addresses the issue of confounding by rigorously measuring the causal effect of length on perceived review quality through a randomized controlled trial where the treatment increases the length of the review without adding useful information.

The papers [19–23] all find that in authors' evaluations of reviews on their own papers, the decision of accept or reject given by the reviewer is highly correlated with evaluation rating given by the authors. However, these prior works do not control for potential confounders. For instance, there may be systematic differences in the true review quality of accept and reject decisions. In our work, we also collect evaluations of reviews by non-authors, which we use to control for these confounders. A related paper is [24] which develops an algorithm to de-bias such author-provided evaluations.

At NeurIPS 2020, the program chairs asked meta-reviewers to rate whether paper reviewers met their expectations [25]. They found that invited reviewers to the conference were not rated any higher than reviewers recruited from among the author pool. Additionally, they found that less experienced reviewers were actually rated slightly higher on review quality than more experienced reviewers. Similarly, a study at the ICML 2020 conference [13] designed a special process to recruit new paper reviewers and asked meta-reviewers to evaluate the review quality from this group and from the standard group of reviewers. They found that their newly recruited and trained reviewers were evaluated as higher quality than reviewers in the standard reviewer pool according to a number of metrics which also included meta-reviewers' evaluations of reviews. Our work does not focus on which reviewers are considered higher quality by meta-reviewers, but rather focuses on the reliability of these evaluations of reviews.

A number of scientific funding agencies collect assessments of peer review quality in the assessment of grant proposals. At Canada's national health research funding agency, committee chairs were asked to evaluate the review quality of grant peer reviewers from 2019 to 2022 [26]. A report from the European Science Foundation on the evaluation of reviews found that such evaluations of review quality were quite common in grant funding agencies—in a survey of 30 funding organizations, they found that over 60% evaluate the quality of all reviews as standard practice [7]. These organizations then use review quality in a number of concrete ways, including to discard reviews deemed low quality and tagging the reviewer with qualifying information for future reference. These policies speak to the importance of assessments of review quality in having real consequences in existing peer review systems of funding agencies. Our work focuses on systematically assessing the reliability of evaluations of review quality.

In medical journals, there is literature going back over two decades on assessing review quality. The study [27] asked editors to evaluate the quality of peer reviews in medical journals and concluded that editors show strong agreement in their evaluations as measured by the intraclass correlation coefficient. Subsequent work [28] tested the efficacy of evaluating reviews by generating a fictitious manuscript with known flaws, obtaining peer reviews of the manuscript and then asking editors to evaluate quality of the peer reviews. They found that evaluation of review quality is somewhat correlated with number of flaws reported by the reviewers, indicating that assessment of review quality may in fact capture some objective qualities that make a review useful. In a cross-sectional study of journals in multiple disciplines, the study [23] analyzed authors' and editors' evaluations of review quality in Elsevier journal reviews from 2014 across medicine, science, and computer science. They found correlation between author satisfaction with the review and whether the review recommended acceptance. Our work studies similar questions on the reliability of evaluating peer review, but in the context of a large Computer Science conference.

A recent paper [29] analyzed whether length of reviews seems to capture review quality. They found a correlation between the length of reviews given to accepted journal articles and the future citations received by these articles, suggesting that review length may be associated with review quality. While it may be the case that longer reviews are sometimes of higher quality than shorter reviews, our work asks whether uselessly elongating reviews can lead to spurious perceptions of higher quality.

## 2 Experimental setup

We note that throughout this paper we use "evaluator/evaluation" to refer to the evaluation of reviews and "review/reviewer" to refer to reviews of papers.

We asked participants at NeurIPS 2022 to evaluate the quality of reviews given on papers at the conference. We recruited four types of evaluators:

1. *Meta-Reviewers*: Asked to evaluate reviews on one paper from their own pool of papers.
2. *Paper Reviewers*: Asked to evaluate other reviewers' reviews on one paper that the participant reviewed for during the conference.
3. *Paper Authors*: Asked to evaluate all reviews on at most 2 of their own submitted papers.
4. *External Reviewers*: Reviewers and meta-reviewers from NeurIPS 2022 with relevant expertise who were asked evaluate all reviews on one paper that they did not handle as part of the conference.

We recruited evaluators on an opt-in basis. Participation was voluntary and no compensation was given. Recruitment occurred after paper submission, during the review period, but before final accept/reject decisions were released. Specifically, we recruited the (meta-)reviewers and authors from July 11 to August 11, 2022 and external reviewers from August 11 to August 25, 2022. The conference used the OpenReview.net conference management platform for the review process. Reviewers, meta-reviewers, and authors were sent a notification about the experiment via this platform, asking if they would like to opt-in. Those who said yes were included. The participants gave written documentation of informed consent and this study was approved by the Carnegie Mellon University Institutional Review Board (IRB).

Given the set of opt-in evaluators, we next chose papers and reviews for them to evaluate in a manner that maximized the amount of overlap in which reviews are evaluated. This was to enable us to then compare the evaluations from multiple evaluators on the same set of reviews. Additionally, in order to ensure that the external reviewers evaluated reviews on relevant papers, we chose papers so that "similarity" between the external reviewers and papers

was high — here, similarity is defined as the similarity between the text of the paper and the text of the reviewers' profile (past papers), which is used in NeurIPS 2022 and various other conferences to assign reviewers to papers in the peer review process.

Overall, we recruited 7,740 evaluators across these 4 types of reviewers who rated 9,870 paper reviews, with a total of 24,638 evaluations completed. Among the participants, there were 493 meta-reviewers, 2,395 paper reviewers, 3,429 paper authors, and 1,423 external reviewers. For context, at the conference there were 824 meta-reviewers, 10,406 reviewers and 9,634 paper submissions in total [30].

Evaluators were provided the review, along with the paper for which the review was written via email. They accessed the reviews through a cloud storage link (personalized for each evaluator) where the reviews and papers were temporarily stored and later deleted. They submitted their evaluations of review quality through a Qualtrics form. The evaluations were completed after paper reviews were completed and shown to authors, but before final decisions on paper acceptance and rejections were made. Evaluators were asked to rate the overall quality of paper reviews on a 7 point scale. Higher ratings correspond to higher rated quality. Additionally, evaluators were asked to evaluate the reviews on the following four criteria:

1. *Understanding*: "The review demonstrates an adequate understanding of the paper."
2. *Coverage*: "The review covers all the required aspects."
3. *Substantiation*: "Evaluations made in the review are well supported."
4. *Constructiveness*: "The review provides constructive feedback to authors."

Evaluators rated each of these criteria on a 5-point Likert scale ranging from –2 (Strongly Disagree) to 2 (Strongly Agree). The evaluation form also contained additional explanation of each of the items: see S1 Appendix for the full questionnaire. We chose these criteria for the questionnaire based on proposed Review Quality Indicators (RQIs) for peer reviews [31,32], additionally tailoring the questions to suit our needs of being concise and relevant to papers in the domain of machine learning.

We describe some basic statistics pertaining to the evaluations. In Fig 1, we show the overall distribution of scores for each type and the distribution of criteria scores. The overall score distribution is symmetric around the median score of 4. The distribution of scores for the criteria are all left-skewed, as evaluators were more likely to give positive scores on these criteria. We further analyze the mapping from criteria scores to overall scores in Sect 3.5.

## 2.1 A randomized control trial

After the first round of evaluations, we conducted a randomized control trial where we manipulated the length of reviews in order to study the impact of review length on perceptions of quality. Specifically, we conducted an experiment where we selected 10 papers such that as many participants as possible had high textual similarity scores (indicating familiarity in the area of the paper) with at least one of the papers. The participants with high similarity scores were drawn from among the external reviewers, giving 458 total evaluators, 334 who served as reviewers and 124 who served as meta-reviewers on other papers at the conference. Importantly, unlike (meta)-reviewers and authors, the participants from this group of external reviewers had not seen the original reviews on these papers, allowing us to manipulate the reviews without their knowledge of the treatment.

For each of the selected papers, we chose one review at random and then manually created a longer version of this review, carefully ensuring that the underlying quality of the review did not improve as we increased the length. We adopted a combination of the following strategies to do so: adding filler text at the beginning of each text box by repeating the text box

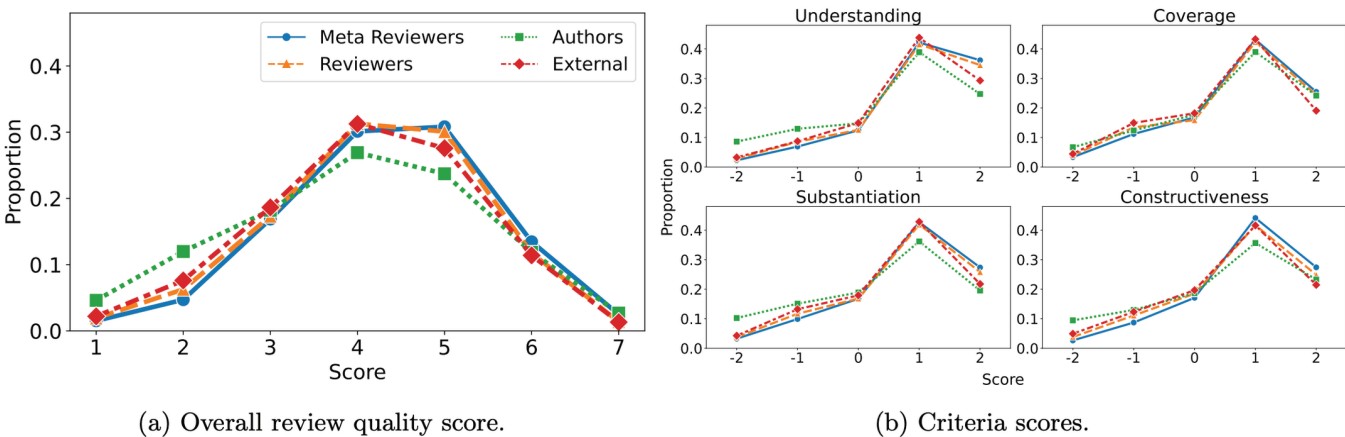

(a) Overall review quality score.

(b) Criteria scores.

**Fig 1. Marginal distribution of scores given to reviews by meta-reviewers, paper reviewers, authors, and external reviewers.**

header as an introductory sentence, repeating the summary in other sections like strengths and weaknesses, writing out the text from multiple-choice questions (Rating, Ethics Flag, Soundness, Presentation, etc.) in the text boxes, replicating the abstract of the paper in the summary box or in the body text of the review. See Fig 2 for an illustration of such an elongation. In S2 Appendix, we give examples of original and elongated reviews used in our experiment that pertain to accepted papers at NeurIPS 2022 which have publicly viewable reviews on OpenReview. As shown in Fig 3, across the 10 reviews the original reviews were roughly 200-300 words long, while the elongated reviews were roughly 600-850 words long. The mean word count of the original reviews was 268 words, compared to a mean of 755 for elongated reviews.

Then, each eligible participant was assigned to exactly one of the experiment papers. Additionally, each participant was assigned uniformly at random to either a "*long*" or "*short*" condition. When asked to evaluate a review for the assigned paper, participants in the *long* group were given the uselessly elongated version of the selected review while participants in the *short* condition were given the original version of the review. Participants were not informed about the specific goal of this additional experiment: we only notified them that the data they contributed would be used to gain insights about the review quality evaluation practice, but did not specifically mention the length confounder. We further discuss the setup of this

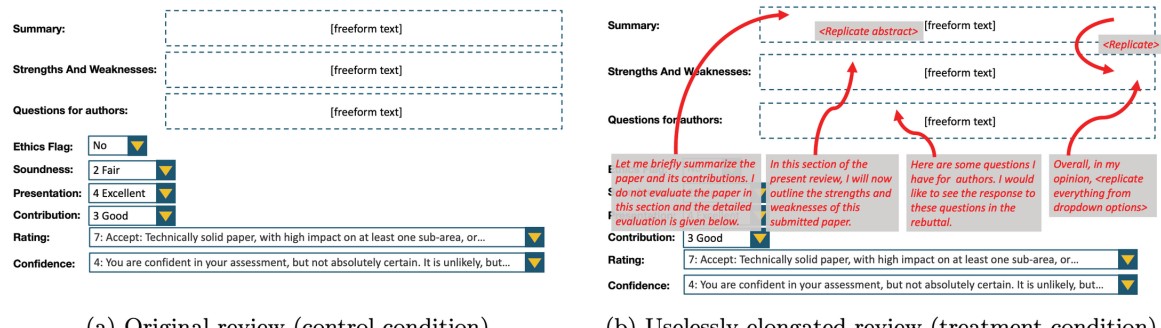

(a) Original review (control condition)

(b) Uselessly elongated review (treatment condition)

**Fig 2. Generation of "uselessly elongated" reviews** by adding unnecessary explanatory text (in red).

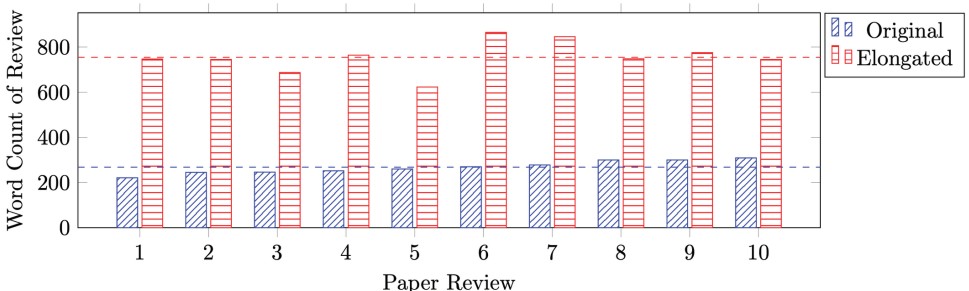

**Fig 3. Word count of the ten original and uselessly elongated reviews.** The word counts include text from the summary, strengths and weaknesses and questions boxes of the paper reviews and exclude the quantitative scores. The mean lengths of original and elongated reviews are shown as dashed lines.

experiment and our analysis in Sect 3.1. We note that the experimental data from the RCT is not used in the rest of our analysis.

# 3 Results

We now present the main results of our analyses on uselessly elongated review bias (Sect 3.1), authors' outcome-induced bias (Sect 3.2), inter-evaluator (dis)agreement 3.3, miscalibration ( Sect3.4), and subjectivity (Sect 3.5).

## 3.1 Uselessly elongated review bias

One concern in evaluations of reviews is that the evaluations may be biased by spurious factors that are not actually indicative of underlying quality, like review length. We hypothesize that evaluators may perceive longer reviewers as better even if they are not of higher quality. In order to rigorously test this hypothesis, we conduct a carefully designed randomized control trial for the effect of "uselessly elongated review bias."

**3.1.1 Methods.** In our experiment, we used 10 reviews written on 10 different papers. For these 10 reviews, we received evaluations from 458 participants. Each of the participants was randomly assigned to either a *short* or *long* condition, meaning they reviewed either the short or long version of a review respectively (as described in Sect 2.1). We then employed the Mann-Whitney U-test to evaluate whether the perceived quality of the 10 selected reviews differs systematically between the *short* and *long* conditions. We compute a Mann-Whitney U-statistic as follows. We take all pairs of evaluations where the two evaluations are of a review on the same paper but one evaluates the short version and the other the long version. There are on average 23 evaluations per paper of the short version of the review and 23 evaluations per paper of the long version giving over 500 pairs of evaluations per paper. For each paper $p \in [10]$ we denote $S_p$ as the set of evaluation scores of the *short* review on the paper and $L_p$ the set of scores of the *long* review on the paper. Then, the test statistic $\tau \in [0, 1]$ is defined as:

$$\tau = \frac{1}{\sum_{p=1}^{10} |L_p||S_p|} \sum_{p=1}^{10} \sum_{x^s \in S_p} \sum_{x^\ell \in L_p} \left( \mathbb{I}(x^\ell > x^s) + 0.5 \, \mathbb{I}(x^\ell = x^s) \right).$$

One can interpret $\tau$ as the probability that a *long* review is scored higher than a *short* review by evaluators, breaking ties in scores at random. Note that under a null hypothesis of no effect, $\tau = 0.5$, so $\tau > 0.5$ indicates a positive bias of review length on quality score and $\tau < 0.5$ indicates negative bias.

To compute confidence intervals for the test statistic $\tau$, we bootstrap reviewers in the *long* and *short* conditions within each review. Specifically, for 20,000 iterations, we independently bootstrap $L_p$ and $S_p$ for each review on each paper $p \in [10]$ and compute the test statistics on the bootstrapped set of reviewers. We then use 2.5 and 97.5 percentiles to construct a 95% Confidence Interval.

To formally test whether reviewers in the *long* and *short* conditions systematically differ in their scores, we apply a two-sided Fisher permutation test. For this, we permute evaluators within each review between the *long* and *short* conditions uniformly at random, ensuring that the number of reviewers in each condition remains the same. We then recompute the value of the test statistic for 20,000 permutations and compare these values with the original value of the test statistic to obtain *p*-values.

**3.1.2 Results.** As shown in Table 1, we find a statistically significant positive impact of length on evaluations of review quality. For both reviewers and meta-reviewers, the uselessly elongated reviews receive higher scores than the original shorter reviews. The effect size for reviewers is similar to the effect size for meta-reviewers. Overall, the mean score for the *long* condition group was 4.29 compared to 3.73 for the *short* condition. As shown in Table 2, we also find a positive effect of length on the criteria scores. In particular, after Holm–Bonferroni correction, results are significant at level 0.05 for all the criteria, with the strongest effect on Coverage. These results suggest that it is possible for a reviewer to spuriously improve perceived quality of their review by adding to their review, even if the additions add no real value.

## 3.2 Authors' outcome-induced bias

One potential source of bias in evaluating review quality that is distinct to authors is bias arising due to the positivity or negativity of a review. A number of past works have documented correlation between author's satisfaction with paper reviews and whether the reviews recommended acceptance [19–23]. We find a similar correlation in our analysis. In Fig 4, we plot the review score given by a paper review against the mean evaluation of review quality given to that review for each type of evaluator. While meta-reviewers, reviewers and external reviewers

**Table 1. Summary of results for the randomized controlled trial testing the effect of uselessly elongated review bias on overall quality score, separated according to the role of the evaluator in the conference.**

| Role | Sample size | $\tau$ | 95% CI | P value | Difference in Means |
|---|---|---|---|---|---|
| Reviewers + Meta-Reviewers | 458 | 0.64 | [0.60, 0.69] | < 0.0001 | 0.56 |
| Reviewers | 334 | 0.65 | [0.59, 0.71] | < 0.0001 | 0.58 |
| Meta-Reviewers | 124 | 0.61 | [0.52, 0.71] | 0.04 | 0.39 |

**Table 2. Summary of results for the randomized controlled trial testing the effect of uselessly elongated review bias on criteria scores. Sample size is 458 for all statistics. Recall that the overall score is on a 7-point scale, while criteria scores are on a 5-point scale.**

| Criteria | $\tau$ | 95% CI | P value | Difference in Means |
|---|---|---|---|---|
| Overall | 0.64 | [0.60, 0.69] | < 0.0001 | 0.56 |
| Understanding | 0.57 | [0.53, 0.62] | 0.04 | 0.25 |
| Coverage | 0.71 | [0.66, 0.76] | < 0.0001 | 0.83 |
| Substantiation | 0.59 | [0.54, 0.64] | 0.001 | 0.31 |
| Constructiveness | 0.6 | [0.55, 0.64] | 0.001 | 0.37 |

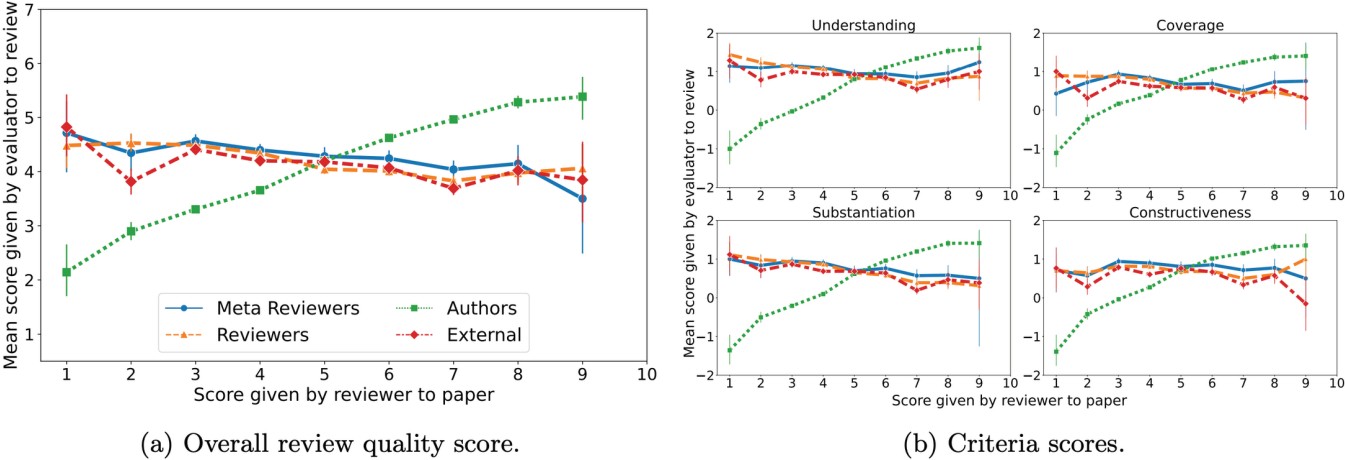

(a) Overall review quality score.

(b) Criteria scores.

**Fig 4. Association between paper review score and review quality evaluation score.** Review score given to a paper by a reviewer (x axis) plotted against the mean evaluation score of that review by evaluators (y axis), for each type of evaluator. Review scores range from 1 (strongest reject) to 10 (strongest accept.) Evaluations of reviews with a score of 10 are omitted from plot due to insufficient sample size ($n = 5$.)

do not show a strong trend in how the positivity of review score correlates with review quality assessments, for authors there is a clear positive trend with reviews recommending strong accepts receiving higher evaluations than reviews recommending strong rejects. This trend holds both for the overall review quality score and for assessments of specific criteria. While this visual suggests such a bias, it does not account for confounding factors, and hence we conduct a formal analysis in this section.

**3.2.1 Methods.** In order to measure the presence of an outcome-induced bias in the evaluations of reviews provided by authors of respective papers, we estimate the effect of receiving a review with a "reject" recommendation versus an "accept" decision on author's evaluations of review quality. We conduct the following non-parametric analysis. We match pairs of evaluations where one evaluation is on a reject review ("weak reject" or below) and the other is on an accept review ("weak accept" or above) based on the following criteria:

1. Evaluation is done by the same author on the same paper.
2. The pair of reviews evaluated have similar length: the longer review is at most $1.5\times$ longer than the shorter.
3. Both reviews have at least 2 evaluations from non-authors and have received a mean overall evaluation score within 1 point of each other from non-authors.

Criteria (i) controls for differences in evaluation of review quality between different authors or papers. Criteria (ii) controls for bias in evaluations due to length of reviews, as found in our RCT described in Sect 3.1. Finally, criteria (iii) controls for underlying differences in the true review quality of the paired reviews. Since, we do not know the ground truth quality of a paper review, the evaluations from non-authors serve as a proxy for true quality. We restrict our analysis to reviews where the non-author evaluators agree in their quality evaluation.

Our matching criteria yields 418 pairs of review evaluations. We then conduct a Mann-Whitney U test on these pairs of evaluations to determine whether accept reviews are likely to receive higher scores than reject reviews. In particular, given the $n = 418$ pairs of scores

$\left\{\left(x_i^{\text{accept}}, x_i^{\text{reject}}\right)\right\}_{i=1}^{n}$, the test statistic $\tau \in [0, 1]$ is computed as:

$$\tau = \frac{1}{n} \sum_{i=1}^{n} \left( \mathbb{I}\left(x_i^{\text{accept}} > x_i^{\text{reject}}\right) + 0.5 \mathbb{I}\left(x_i^{\text{accept}} = x_i^{\text{reject}}\right) \right).$$

One can interpret the test statistic $\tau$ as the probability that an accept rating is scored higher than a reject rating by authors, breaking ties in scores at random. We run a two sided Fisher permutation test with 20,000 simulations to determine a *p*-value of the test statistic. The 95% confidence intervals are bootstrapped with 20,000 simulations.

**3.2.2 Results.** We find a treatment effect of $\tau = 0.82$ with a *p*-value of < 0.0001 in the overall quality scores, indicating that authors are positively biased towards reviews recommending accept over reviews recommending reject. Additionally, on average accept reviews received scores that were 1.406 points higher (on the 7 point evaluation scale) than reject reviews. We additionally test for differences in the criteria scores between the matched pairs of accept reviews and reject reviews. As shown in Table 3, we find a positive bias towards accept reviews on the Understanding, Coverage, Substantiation, and Constructiveness criteria respectively. These results are all statistically significant at a level of 0.05 after Holm-Bonferroni correction. The criteria scores were roughly 1 point higher on the 5-point review scale for accept reviews than Reject reviews. This indicates that authors' positive bias towards reviews recommending accept manifests in criteria scores as well as overall scores. We note that authors did not have any explicit incentive in our experiment to rate accept reviews higher than reject reviews: there were no repercussions to paper reviewers for receiving positive or negative evaluation scores for their paper reviews nor for the acceptance decisions. Nonetheless, authors seemed to display an inherent bias towards reviews that were more positive towards their work. These results suggest that caution must be taken when asking authors to evaluate reviews on their own papers.

## 3.3 Inter-evaluator (dis)agreement

One measure of the evaluation reliability is the consistency of scores. Consistency by itself is not sufficient for a useful evaluation process, for example, consistency is high if most evaluators simply give the median score out of laziness, but these evaluations are not useful. Nonetheless, consistency is one factor in evaluating reliability of evaluations, as we would generally like to obtain similar evaluations of review quality if we ask multiple people.

With this motivation, we follow the methods of [33] in their analysis of the reviews of papers (*not* evaluations of reviews) in the peer-review process of the NeurIPS 2016 conference. The NeurIPS 2016 conference asked reviewers to evaluate reviews on four criteria (but did not ask for an overall score). The analysis [33] computes the rate of agreement between

**Table 3. Summary of results for Mann-Whitney U test of authors' bias towards reviews recommending accept compared to reviews recommending reject (on *n* = 418 pairs of reviews).**

| Criteria | $\tau$ | 95% CI | P value | Difference in Means |
|---|---|---|---|---|
| Overall | 0.82 | $[0.79, 0.85]$ | < 0.0001 | 1.41 |
| Understanding | 0.78 | $[0.75, 0.81]$ | < 0.0001 | 1.12 |
| Coverage | 0.76 | $[0.72, 0.79]$ | < 0.0001 | 0.97 |
| Substantiation | 0.80 | $[0.76, 0.83]$ | < 0.0001 | 1.28 |
| Constructiveness | 0.77 | $[0.74, 0.80]$ | < 0.0001 | 1.15 |

reviews provided by a pair of reviewers on a pair of papers that they both review. In this manner, we compare the amount of agreement in reviews of papers (in NeurIPS 2016) with the amount of agreement in evaluations of reviews (in NeurIPS 2022).

**3.3.1 Methods.** We compute the inter-evaluator (dis-)agreement following [33]. Consider any individual criterion or the overall score. Take any pair of evaluators and any pair of reviews that receives an evaluation from both evaluators. We say the pair of evaluators agrees on this pair of reviews if both score the same review higher than the other; we say that this pair disagrees if the review scored higher by one evaluator is scored lower by the other. Ties are discarded. We then compute the total number of agreements and disagreements. The total sample size (number of quadruples of paired review scores) in our calculations was $n_{\text{overall}} = 25,346$ for the overall score and $n_{\text{understanding}} = 18,658, n_{\text{coverage}} = 18,193, n_{\text{substantiation}} = 19,614, n_{\text{constructiveness}} = 19,870$ for each of the criteria scores. We show disagreement rates along with 95% confidence intervals in Fig 5. We note that a random baseline for the agreement rate if scores are drawn independently at random for each evaluation of a review (from any marginal distribution) is 0.5.

**3.3.2 Results.** For the overall score, 29% of pairs of evaluations were ties, while for each of the criteria scores 37% to 40% of pairs were ties. In comparison, in the reviews of papers in NeurIPS 2016 [33], 35%–40% of pairs of criteria scores were tied. We now plot the rates of disagreements for the evaluations of NeurIPS 2022 reviews in Fig 5. The disagreement rates for both overall quality score and the criteria scores are approximately 0.3 on all criteria. In comparison, the same inter-evaluator disagreement statistic for reviews of papers in the NeurIPS 2016 [33], is in the rage of 0.25 to 0.3. While the domains are different, these results suggest that evaluations of reviews and evaluations of papers have similar agreement rates. In fact, an experiment at NeurIPS 2021 found that the rate of disagreement between co-authors of multiple jointly authored papers about the contribution of their own papers is 0.32, and that between authors of papers and the review process is 0.34 [34]. These disagreement rates are similar to what we found here for reviews of reviews.

## 3.4 Miscalibration

Another issue in peer review of papers is evaluator miscalibration, that is the tendency for evaluators to exhibit idiosyncrasies such as giving especially lenient or harsh reviews [35–37]. In this section, we investigate whether the problem of miscalibration manifests itself in evaluating review quality.

**3.4.1 Methods.** In order to estimate the degree of miscalibration, we fit a simple model that assume linear miscalibration in scores for each reviewer [38]. This allows for comparison to prior work in estimating miscalibration in paper review, where the same model of evaluation scores is employed. Specifically, we follow the methods of [38], modeling the evaluation scores as a linear combination of objective quality, evaluator bias and per-evaluation

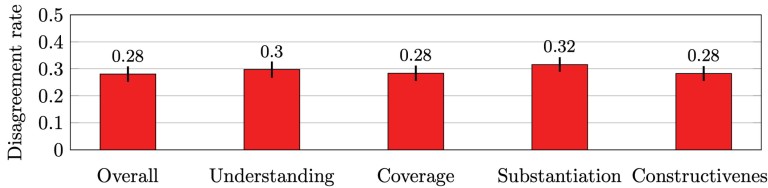

**Fig 5. Inter-evaluator disagreement rates given to reviews.**

idiosyncrasy. The model assumes that the overall quality score given by evaluator $j$ to review $i$, denoted as $y_{ij}$, is given by

$$y_{ij} = f_i + b_j + \epsilon_{i,j},$$

where

- $f_i \sim \mathcal{N}(\mu, \alpha_f)$ is an assumed "objective quality" of review $i$ in the model, drawn from a normal distribution with mean $\mu$ and variance $\alpha_f$;
- $b_j \sim \mathcal{N}(0, \alpha_{b,g})$ is an "evaluator offset" capturing miscalibration of evaluator $j$. In order to capture differences in distributions of the four types of evaluators (meta-reviewers, reviewers, authors, and opt-in reviewers) we model the evaluator offset as a separate per-type normal distribution with mean 0 and variance $\alpha_{b,t}$ for $t \in \{1, 2, 3, 4\}$;
- $\epsilon_{i,j} \sim \mathcal{N}(0, \sigma^2)$ is the idiosyncrasy associated to this specific evaluation of review $i$ by evaluator $j$.

This model is a Gaussian process with 6 variance hyperparameters to learn from evaluation data. We fit the parameters using maximum likelihood estimation. (We use Gaussian Process Regression maximum likelihood estimation implemented in the python package GPy.) We are particularly interested in the $\alpha_{b,g}$ parameter, the estimated variance of evaluator offsets for each reviewer type. Intuitively, $\alpha_{b,g}$ captures the degree of miscalibration for each type, with a larger value indicating that evaluators of that type are more likely to be miscalibrated.

**3.4.2 Results.** In Table 4, we enumerate the values of the fit parameters, normalized by variance of objective quality scores $\alpha_f$. First, observe that the (normalized) variance of author offset of 0.780 is much higher than the variance of evaluator offset for other types of approximately 0.45, suggesting that authors may be more likely to be miscalibrated than other types of reviewers. Second, let us compare with the miscalibration in the NeurIPS 2014 reviews of papers [38]. As mentioned earlier, we use the same model as that used in [38] to enable a direct comparison. The only difference is that the 2014 analysis had a single $\alpha_b$ term whereas we have a separate term for each evaluator type. For the NeurIPS 2014 reviews of papers, it was found that $\alpha_f = 1.28$, $\alpha_b/\alpha_f = 0.19$ and $\sigma^2/\alpha_f = 1.01$. This suggests that under the linear model of score generation, miscalibration in evaluating review quality may be at least as high as compared to evaluating paper quality.

## 3.5 Subjectivity (commensuration bias)

A frequent concern in peer review is subjectivity of reviewers. In the context of paper review, one source of subjectivity arises from reviewers having differing opinions about the relative importance of various criteria in determining overall quality of a paper, a phenomenon referred to as *"commensuration bias"* [39]. For example, some reviewers may consider novelty of a paper more important towards overall quality whereas others may consider rigor more

**Table 4. Fit parameters of linear calibration model.**

| | |
|---|---|
| $\alpha_f$ (Objective Quality Variance) | 0.581 |
| $\alpha_{b,1}/\alpha_f$ (Meta-Reviewer Offset Variance) | 0.458 |
| $\alpha_{b,2}/\alpha_f$ (Reviewer Offset Variance) | 0.432 |
| $\alpha_{b,3}/\alpha_f$ (Author Offset Variance) | 0.780 |
| $\alpha_{b,4}/\alpha_f$ (External Offset Variance) | 0.441 |
| $\sigma^2/\alpha_f$ (Subjective Score Variance) | 1.467 |

important. In our context of evaluating reviews, we asked evaluators to assess the quality of reviews on four specific criteria—understanding of the paper, coverage of required aspects of a review, substantiation with evidence, and constructiveness of the feedback. The overall score given by the evaluator then depends on how the evaluator maps these individual criteria to an overall quality score, and such a commensuration bias can result in arbitrariness in the evaluation process.

**3.5.1 Methods.** Previous research has proposed learning a function that maps criteria scores to overall scores from the review data [40]. At a high level, this learned function is one that best fits the data while respecting monotonicity so that the function is consistent (that is, an improvement in any one criterion holding other criteria constant should not decrease the overall evaluation). We can obtain one measure of the degree of commensuration bias in our evaluation process by computing the loss of this aggregate function learned from the evaluation data, where the loss is defined as the absolute difference between this aggregate function and the overall scores given by evaluators (averaged across all evaluations). Higher loss indicates that there is more variability in how evaluators map criteria to overall scores, suggesting higher commensuration bias. Following the theory developed in [40], we choose the $L(1,1)$ norm as our loss function. We note that higher loss could also arise due to unobserved differences between reviews with the same criteria evaluation scores in addition to commensuration bias. We are not aware of methods that try to control for such latent sources of subjectivity and commensuration bias simultaneously.

In our approach, we learn a single function that is common to all the types of evaluators. An alternative approach would be to learn a separate function for each type. In order to evaluate the usefulness of this alternative approach, we randomly partition evaluation scores into a 75% – 25% train-test split. We then fit a combined-evaluator type function on the training data and per-evaluator type functions on the training data to minimize $L(1,1)$ loss. We evaluate the two approaches on the test data to obtain estimated test loss. To predict on criteria scores that were not present in the train data, we solve a convex optimization problem to minimize $L(1,1)$ loss subject to monotonicity constraints with respect to the function learned on the train data and other points in the test data. Repeating this procedure 5 times, we find that the combined type function achieves a train loss of 0.456 and a test loss of 0.457, while the per-type functions achieves a train loss of 0.448 and a test loss of 0.465. This indicates that estimating different functions per-type does not improve model quality, so we continue to use the combined-evaluator type model.

**3.5.2 Results.** Comparing overall scores given by evaluators to the scores assigned by the learned mapping from criteria scores to overall scores, we find evaluators had a mean loss of approximately 0.45. As a point of comparison, we also evaluate subjectivity (commensuration bias) in NeurIPS 2022 *paper* review data. We employ the same approach for the reviews on papers as we did for evaluations of reviews: we estimate a function mapping criteria scores to overall scores on the 33,371 reviews for papers in NeurIPS 2022 and compute the mean $L1$ loss. We note that the overall scores in the reviews of papers at NeurIPS 2022 used a 10 point review scale, whereas our evaluations of reviews used a 7 point review scale. We thus renormalize the loss by 6/9 (assuming a linear mapping from the 10 point scale to the 7 point scale). We find that the loss on reviews of papers 0.402. While the criteria are different in the review of papers and evaluations of reviews, this result suggests that the degree of subjectivity (commensuration bias) is similar in paper review and in evaluating review quality at NeurIPS 2022.

## 4 Discussion and limitations

In this work, we analyze the reliability of peer reviewing peer reviews. We find that many problems that exist in peer reviews of papers—inconsistencies, biases, miscalibration, subjectivity—also exist in peer reviews of peer reviews. In particular, while reviews of reviews may be useful in designing better incentives for high-quality reviewing and to measure effects of policy choices in peer review, considerable care must be taken when interpreting reviews of reviews as a sign of review quality.

### 4.1 Limitations

Our study has several limitations. First, participants in the experiment knew they were providing evaluations for an experiment, which may result in "Hawthorne" effects. Relatedly, it may be that evaluators behave differently when evaluations of reviews are used for downstream decisions with actual consequences for reviewers such as to give out paper awards. For example, it is possible that evaluators put in more effort when their reviews of reviews have concrete consequences. Second, our study was conducted on an opt-in basis and was not compulsory. There may be selection bias in which authors, reviewers, and meta-reviewers chose to participate in evaluating reviews. In many of our experiments, we separately analyze the four types of evaluators, which accounts for selection bias in which types decided to opt-in, but there still may be selection biases within each type. Third, a limitation in the length experiment is that we were only able to use reviewers/meta-reviewers who did not themselves review the paper, since original reviewers had seen the actual reviews. While these evaluators were provided the associated paper, it will be of interest to test effect of length on evaluations of review quality by other reviewers or authors of a paper, who may be more familiar with the paper content. Lastly, in comparison to reviews of papers (in particular, on subjectivity and miscalibration), the review scales used are different — we use a 7-point rating scale while paper reviews at NeurIPS (to which we compare as a baseline) are evaluated on a 10-point rating scale. While we re-normalize so that metrics from different domains share the same scale, there may be other effects in the use of different scales that are not accounted for.

There is one prominent problem which exists in reviews of papers which we are unable to study in the context of reviewing reviews—dishonest behavior. One form of dishonest behavior is that of "lone wolf" dishonesty in which reviewers, who are also authors of some submitted papers, deliberately manipulate the reviews they provide to increase the chances of their own papers being accepted [41–43]. A second form of dishonest behavior that has gained significant importance recently is that of collusion rings [44–46]. Here, a group of reviewers make a pact according to which they try to get assigned each others' papers for review, and provide positive reviews to each other. In our study, the participants had no incentives for dishonesty since the review-quality evaluations had no downstream consequences in terms of paper acceptances. However, it is not hard to envisage that if the stakes of reviewing reviews become high (e.g., reviewer awards become important or even necessary for promotion) dishonest behavior may also be a problem in reviewing of reviews.

### 4.2 Open problems

These limitations notwithstanding, this study has implications for the use of evaluations of reviews in improving the scientific peer-review process. In particular, our results suggest that evaluations of review quality are rife with issues like biases, inconsistency, subjectivity, and miscalibration. This indicates that we need more reliable approaches to evaluate the quality of reviews. For example, it may be helpful to consider some semi-automated or fully automated

approaches to evaluation of review quality. In the applications of designing incentive mechanisms and measuring impacts of interventions in peer review, our results suggest that care needs to be taken in using human evaluations of review quality for these uses.

Some past works on incentivizing high quality paper review content ([2,3]) have assumed that evaluators of review quality report "true quality." Our results suggest instead that evaluators provide scores rife with biases and noise. Hence, incentive mechanisms need to account for these sources of noise and bias in order to fairly reward high quality review and penalize low quality review. In particular, the "uselessly elongated review bias" may create problems for the design of incentives for high quality review. On the one hand, our work suggests that reviewers who would like to be rewarded for higher quality review may be able to uselessly lengthen their reviews in order to be perceived as higher quality. On the other hand, longer reviews may genuinely be higher quality if a reviewer has completed a more detailed and thoughtful evaluation of a paper. Hence, an incentive designer needs to carefully account for review length, which may constitute a cheap (spurious) signal or a genuine signal of quality.

The issues in evaluating review quality also create issues when measuring the impact of an experiment in peer review. For instance, there is much recent interest in using large-language models (LLMs) for reviewing papers [1,15,47,Section 9.6]. One recent study [15] generated reviews for a set of papers using the GPT-4 model and then asked authors of these papers to compare the quality of the model-generated reviews to human-written reviews. They found that LLM-generated reviews were rated as more helpful than some human-generated reviews. Our results indicate that these experiments, which use author's evaluations of reviews on their own papers, should take into account any bias stemming from the positivity or negativity of reviews given. Furthermore, if the LLM was writing uselessly longer reviews (e.g., the LLM adds more filler sentences), then uselessly elongated review bias could lead to false positive conclusions in this study. Thus, it is important to check for potential length bias when interpreting the effect of using an LLM to generate reviews.

In conclusion, our work pinpoints a number of specific pitfalls in evaluating review quality, which may negatively impact downstream applications that use these evaluations. It is an important open problem to address these concerns either by designing better methods for evaluating review quality or by taking into account for sources of bias and inconsistency in reviews in downstream applications.

## Supporting information

**S1 Appendix: Questionnaire for participants**
(PDF)

**S2 Appendix: Original and extended reviews**
(PDF)

## Acknowledgments

We are greatly indebted to the participants of this experiment for providing evaluations of reviews, thereby helping understand the promises and challenges of evaluating review quality, and consequently also shedding light on the design of incentives and experiments in peer review.

## Author contributions

**Conceptualization:** Ivan Stelmakh, Nihar B. Shah.

**Formal analysis:** Alexander Goldberg, Ivan Stelmakh, Nihar B. Shah.

**Investigation:** Ivan Stelmakh.

**Methodology:** Alexander Goldberg, Ivan Stelmakh, Nihar B. Shah.

**Project administration:** Nihar B. Shah.

**Resources:** Kyunghyun Cho, Alice Oh, Alekh Agarwal, Danielle Belgrave.

**Supervision:** Nihar B. Shah.

**Writing – original draft:** Alexander Goldberg, Nihar B. Shah.

**Writing – review & editing:** Alexander Goldberg, Nihar B. Shah.

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
