## [Decision Letter · Decision Letter 0]

29 May 2024

PONE-D-24-02296

Peer reviews of peer reviews: A randomized controlled trial and other experiments

PLOS ONE

Dear Dr. Goldberg,

Thank you for submitting your manuscript to PLOS ONE. After careful consideration, we feel that it has merit but does not fully meet PLOS ONE’s publication criteria as it currently stands. Therefore, we invite you to submit a revised version of the manuscript that addresses the points raised during the review process.

**Please follow the comments of the reviewer.**

We look forward to receiving your revised manuscript.

Kind regards,

Guglielmo Campus, Ph.D DDS

Academic Editor

PLOS ONE

Journal Requirements:

Additional Editor Comments:

Dear Authors,

The paper was revised and before to move the paper further.it is needing that you follow the comments of the reviewer.

Reviewers' comments:

Reviewer's Responses to Questions

**Comments to the Author**

1. Is the manuscript technically sound, and do the data support the conclusions?

Reviewer #1: Yes

Reviewer #2: Yes

2. Has the statistical analysis been performed appropriately and rigorously? 

Reviewer #1: Yes

Reviewer #2: Yes

3. Have the authors made all data underlying the findings in their manuscript fully available?

Reviewer #1: No

Reviewer #2: No

4. Is the manuscript presented in an intelligible fashion and written in standard English?

Reviewer #1: Yes

Reviewer #2: Yes

5. Review Comments to the Author

Reviewer #1: This paper investigates biases and noise in assessing the quality of peer reviews. It is a very nice contribution to the literature, with arguably the centerpiece being a clever (and funny) randomized controlled trial on review length. I think the paper is in great shape and have relatively minor comments:

- Section 3.1.1. Methods repeats some information that's already in section 2.1.

- The number of evaluators and evaluations recruited is impressive, and given that there were no incentives, surprising. It would help to add some detail on the recruitment pipeline -- was the questionnaire and consent step implemented during the course of the normal review process? And the RCT probably happened later? Or perhaps this information is already there and I missed it?

- It would help giving a little more detail on the choice/setup of the Mann-Whitney tests: for example, why are we calculating the confidence intervals and p-values separately, and using different number of iterations? And why the conventional Mann-Whitney or Wilcoxan rank-sum tests not appropriate here?

- I was a confused in section 3.2.1. criterion (iii). This suggests you wanted "consensus" papers? And how does this information from non-authors enter the test statistic? I expected some quantity like "Author_score - mean(Non-author_score)" but didn't see it.

- For the 3.5 Subjectivity analysis, commensuration bias, or in general differences in mapping are certainly one type of subjectivity, but other types exist too. In fact, it wasn't clear why this approach, estimating the loss, was more information than looking at the linear model of section 3.4.1. Why couldn't we take the variance of b_j and \epsilon_i,j as the more natural definition of "subjectivity"? Furthermore, I don't think it's guaranteed that a final score is the result of *only* the criterion scores. What if the final score is some mapping from the criterion scores *and some unobserved criterion*, and let's say there is some subjectivity around that one -- then we might have a consistent mapping function but still substantial loss.

- It would help the reader contextualize the results if the abstract noted that responses weren't incentivized.

Data availability

- If I understood correctly, the data availability statement is of the "Available upon request" type, but at the same time notes all the reasons why it can't be available. There even seems to be a promise to the participants that "The data will not be distributed," perhaps not even with anonymization. I really hope *something* can be made available publicly, as plenty of research has revealed that "available upon request" generally means "not available" and because I'm a big fan of the paper. It seems that thorough anonymization is quite possible here so in theory it should be possible to share the anonymized data. However if the promise made to participants is that nothing would be shared no matter what, then I think there's a challenge for PLOS policies.

Reviewer #2: Elucidating factors surrounding the quality of reviews of scientific work is important. The authors performed a randomized controlled trial to determine whether review length influences the subsequently rated quality of the reviews. The manuscript is well-written and the objectives of the study are addressed in the results and discussion sections. However, the in-text references throughout the paper (notations like KBY10 instead of Vancouver style numbering) and references at the end of the manuscript are not formatted according to PLoS guidelines. Additionally, modification of the RQI for the present study should be better explained. Please find below detailed comments according to specific sections of the manuscript, thank you.

Introduction

1. Please follow the journal’s Vancouver style of formatting references: https://journals.plos.org/plosone/s/submission-guidelines#loc-references.

Methods

1. P. 4, lines 132-133 – How were the evaluators recruited? Were the email addresses obtained from the conference organizer or was there some other way? Similarly, how did the evaluators who agreed to participate receive the reviews?

2. P. 5, lines 162-164 – Since the authors chose certain questions from the RQI, and tailored the questions intended for machine learning, did the authors consider it a limitation that re-validation of the modified RQI would be warranted? The RQI was intended for medicine, so are the modified questions of the RQI questionnaire in this study measuring what they intend to measure?

6. PLOS authors have the option to publish the peer review history of their article (what does this mean?). If published, this will include your full peer review and any attached files.

Reviewer #1: No

Reviewer #2: **Yes: **Shelly Melissa Pranic

---

## [Decision Letter · Decision Letter 1]

19 Feb 2025

Peer reviews of peer reviews: A randomized controlled trial and other experiments

PONE-D-24-02296R1

Dear Authors

We’re pleased to inform you that your manuscript has been judged scientifically suitable for publication and will be formally accepted for publication once it meets all outstanding technical requirements.

Kind regards,

Guglielmo Campus, Ph.D DDS

Academic Editor

PLOS ONE

Additional Editor Comments (optional):

Reviewers' comments:

Reviewer's Responses to Questions

**Comments to the Author**

1. If the authors have adequately addressed your comments raised in a previous round of review and you feel that this manuscript is now acceptable for publication, you may indicate that here to bypass the “Comments to the Author” section, enter your conflict of interest statement in the “Confidential to Editor” section, and submit your "Accept" recommendation.

Reviewer #2: All comments have been addressed

2. Is the manuscript technically sound, and do the data support the conclusions?

Reviewer #2: Yes

3. Has the statistical analysis been performed appropriately and rigorously? 

Reviewer #2: Yes

4. Have the authors made all data underlying the findings in their manuscript fully available?

Reviewer #2: No

5. Is the manuscript presented in an intelligible fashion and written in standard English?

Reviewer #2: Yes

6. Review Comments to the Author

Reviewer #2: The authors have fully addressed my comments and suggestions in the current version of the manuscript. Thank you.

7. PLOS authors have the option to publish the peer review history of their article (what does this mean?). If published, this will include your full peer review and any attached files.

Reviewer #2: **Yes: **Shelly Melissa Pranic

---

## [Editor Report · Acceptance letter]

PONE-D-24-02296R1

PLOS ONE

Dear Dr. Goldberg,

I'm pleased to inform you that your manuscript has been deemed suitable for publication in PLOS ONE. Congratulations! Your manuscript is now being handed over to our production team.

Kind regards,

on behalf of

Prof. Dr. Guglielmo Campus

Academic Editor

PLOS ONE